# Genome-Wide Identification of *WRKY* Gene Family and Functional Characterization of *CcWRKY25* in *Capsicum chinense*

**DOI:** 10.3390/ijms241411389

**Published:** 2023-07-13

**Authors:** Liping Zhang, Dan Wu, Wei Zhang, Huangying Shu, Peixia Sun, Chuang Huang, Qin Deng, Zhiwei Wang, Shanhan Cheng

**Affiliations:** 1Key Laboratory for Quality Regulation of Tropical Horticultural Crops of Hainan Province, Sanya Nanfan Research Institute, Hainan University, Sanya 572000, China; zlp602900665@163.com (L.Z.); cocoapuff_93@163.com (D.W.); zhang914wei@163.com (W.Z.); hnhyshu@163.com (H.S.); denggine@163.com (Q.D.); wangzhiwei@hainanu.edu.cn (Z.W.); 2Key Laboratory for Quality Regulation of Tropical Horticultural Crops of Hainan Province, School of Horticulture, Hainan University, Haikou 570228, China; sunshine06261221@163.com (P.S.); chaung11@163.com (C.H.)

**Keywords:** pepper, *Capsicum chinense*, *WRKY* gene family, capsaicinoid biosynthesis, *CcWRKY25*

## Abstract

Pepper is renowned worldwide for its distinctive spicy flavor. While the gene expression characteristics of the capsaicinoid biosynthesis pathway have been extensively studied, there are already a few reports regarding transcriptional regulation in capsaicin biosynthesis. In this study, 73 *WRKYs* were identified in the genome of *Capsicum chinense*, and their physicochemical traits, DNA, and protein sequence characteristics were found to be complex. Combining RNA-seq and qRT-PCR data, the *WRKY* transcription factor *CA06g13580*, which was associated with the accumulation tendency of capsaicinoid, was screened and named *CcWRKY25*. *CcWRKY25* was highly expressed in the placenta of spicy peppers. The heterologous expression of *CcWRKY25* in *Arabidopsis* promoted the expression of genes *PAL*, *4CL1*, *4CL2*, *4CL3*, *CCR*, and *CCoAOMT* and led to the accumulation of lignin and flavonoids. Furthermore, the expression of the capsaicinoid biosynthesis pathway genes (CBGs) *pAMT*, *AT3*, and *KAS* was significantly reduced in *CcWRKY25*-silenced pepper plants, resulting in a decrease in the amount of capsaicin. However, there was no noticeable difference in lignin accumulation. The findings suggested that *CcWRKY25* could be involved in regulating capsaicinoid synthesis by promoting the expression of genes upstream of the phenylpropanoid pathway and inhibiting CBGs’ expression. Moreover, the results highlighted the role of *CcWRKY25* in controlling the pungency of pepper and suggested that the competitive relationship between lignin and capsaicin could also regulate the spiciness of the pepper.

## 1. Introduction

*Capsicum* spp. is an annual or perennial plant of the genus in the family Solanaceae and comprises five cultivated species, including *C. annuum*, *C. chinense*, *C. frutescens*, *C. baccatum*, and *C. pubescens* [1]. Pepper fruit is rich in vitamins, minerals, carotenoids, and other nutrients, as well as having a distinctive spicy flavor, making it useful for various purposes such as fresh food, condiments, insect and microbial resistance, medicine, personal protection, and other fields [2,3,4,5].

Capsaicin is the primary compound responsible for the spicy taste in peppers; it is synthesized in the placenta of the fruit via the phenylpropanoid pathway (PP) and the branched-chain fatty-acid pathway (BCFAP) [6]. Both of these pathways are shared with other metabolic processes, such as the production of lignin, flavones, tannins, and alkaloids. The activity of enzymes involved in PP and each branch of BCFAP can influence the direction and rate of capsaicin metabolism [7]. Approximately 50 genes, including *PAL*, *C4H*, and *4CL* involved in PP and *BCAT*, *KAS*, and *ACS* involved in BCFAP, as well as *pAMT*, *AT3*, *CCR*, and *CAD*, have been identified as being linked to the formation of the spiciness [8,9]. Currently, the transcription factors that regulate capsaicinoid synthesis include the *MYB*, *AP2/ERF*, and *WRKY* gene families. Keyhaninejad et al. demonstrated that the expression patterns of *Jerf* and *Erf* in the *AP2/ERF* family largely coincided with the accumulation of capsaicin, indicating that they may be engaged in the control of capsaicinoid biosynthesis [10]. Arce-Rodriguez et al. first identified that the *CaMYR31* transcription factor was a regulator of capsaicinoid biosynthesis [11]. Via VIGS technology, it was found that *CaMYB31* silencing decreased capsaicinoid content and the transcription of CBGs, further supporting its role in the capsaicinoid biosynthesis pathway [12]. Zhu et al. identified an *MYB* transcription factor specific to the Solanaceae family that was homologous to *MYB31* and encoded by the *Pun3* locus, which was particularly expressed in the placenta and synergistically expressed with structural genes to promote capsaicinoid synthesis [13]. In-depth research revealed that *WRKY9* indirectly affected capsaicinoid synthesis by binding to the promoter W-box of *MYB31* [13]. However, due to the complexity of the phenylpropanoid pathway and the modulation of each branch’s metabolism, there could be additional transcriptional regulations of the capsaicinoid biosynthesis pathway yet to be discovered.

*WRKY*, one of the most prominent transcription factor families in higher plants, has been identified in *Arabidopsis* [14], *Oryza sativa* [15], *Solanum lycopersicum* [16], and *Nicotiana tabacum* [17], with 74, 109, 81, and 164, respectively. The *WRKY* gene family has a WRKY protein domain with a length of 60–70 amino acids, consisting of a highly conserved heptapeptide WRKYGQK motif at the N-terminus and a zinc finger structure of C-X4-5-C-X22-23-H-X-H (C_2_H_2_) or C-X5-7-C-X23-28-H-X-C (C_2_HC). The WRKYGQK motif usually recognizes the DNA-binding promoter element W-Box (C) (T) TGAC (T/C) in the upstream region of the inducible gene, promoting sequence-specific binding and activation with the W-Box [18,19]. Based on the type of zinc finger motif and the number of WRKY domains, WRKY proteins are classified into three groups: members of Group I contain two WRKY structural domains and one C_2_H_2_-type zinc finger structure; members of Group II only contain one WRKY domain and one C_2_H_2_-type zinc finger structure; members of Group III also have only one WRKY structural domain, but the zinc finger structure type is C_2_HC. Within Group II, members are further divided into subgroups IIa, IIb, IIc, IId, and IIe, based on their amino acid sequences. *WRKYs* are essential transcription factors with several roles in regulating biotic and abiotic stress [20], plant senescence [21], and morphogenesis [22]. In addition to these physiological processes, *WRKYs* are also engaged in secondary metabolic processes, such as the biosynthesis of lignin [23], terpenoids [24], flavonoids [25], and alkaloids [13], which are crucial for plant growth and environmental adaptation. Therefore, given the competitive synthesis interaction between capsaicin, lignin, and flavonoids, the role of *WRKY* in the regulation of spiciness remains to be confirmed.

*C. chinense* is known to contain significantly more capsaicinoid than other cultivated pepper species [26]; its genome has been published [27]. However, the *WRKY* gene family has not been extensively characterized in *C. chinense*. In this research, the *WRKY* gene family was identified from *C. chinense* by bioinformatics. Based on the RNA-seq of *C. chinense* placenta at 10–50 days after anthesis, in conjunction with the transcriptomic datasets of pepper root, stem, leaf, bud, flower, and fruit, candidate *WRKY* genes that might be related to pungency were identified. We cloned the identified gene and performed functional identification by qRT-PCR, *Arabidopsis* genetic transformation, and VIGS, which would serve as a theoretical foundation for elucidating the transcriptional regulatory mechanism of capsaicinoid biosynthesis.

## 2. Results

### 2.1. Identification and Sequence Analysis of Genome-Wide WRKYs in C. chinense

A total of 73 *WRKY* transcription factors were obtained in the *C. chinense* genome and designated *CcWRKY1*-*CcWRKY73*. There were significant differences in the physicochemical properties of the members of the *WRKY* gene family, including the length of amino acid from 160 aa (*CcWRKY33*) to 760 aa (*CcWRKY54*), the molecular weight of protein from 18.6 to 82.6 kDa, and the isoelectric point from 5.52 (*CcWRKY71*) to 10.14 (*CcWRKY49*). It was estimated that the CcWRKY proteins were all unstable and the hydrophilic proteins were according to the instability index and grand average of hydrophilicity. Moreover, 73 *CcWRKYs* were located in the nucleus (Appendix A).

### 2.2. Multiple Sequence Alignment and Phylogenetic Analysis of CcWRKY Gene Family

The 73 CcWRKYs could be classified into Groups I, II, and III transcription factors, with the numbers 16, 47, and 10, respectively, according to the phylogenetic tree of the CcWRKY and AtWRKY proteins. Group II could be further divided into five subgroups of IIa, IIb, IIc, IId, and IIe, each containing 4, 8, 16, 5, and 14 members (Figure 1). Protein sequence alignment analysis revealed that most CcWRKY proteins contained intact conserved domains “WRKYGQK”, with only a small number of amino acids undergoing specific mutations and evolution (Figure 2). All of them belonged to Group I and had complete N- and C-terminal domains, whereas Group IIe’s *CcWRKY49*, *CcWRKY47*, and *CcWRKY24* and Group III’s *CcWRKY63* lost their zinc finger motifs. Despite the existence of the “WRKYGQK” sequence in the *WRKY* gene family, there were six *WRKY* transcription factors with variant WRKY domains among the *CcWRKY* members, of which WRKYGKK (three, all Group IIe) was the more common variant, along with WKRYGHK (*CcWRKY26*), WKRYGMK (*CcWRKY44*), and WKRYGQT (*CcWRKY31*).

### 2.3. Analysis of Gene Structure, Conserved Motifs, and Structural Domains of the CcWRKY Gene Family

For the purpose of gaining further insight into the *CcWRKY* information, the phylogenetic tree, gene structure, and motif were built using TBtools (v1.045). The evolutionary relationships showed that the *CcWRKY* gene family was generally in line with the findings of the phylogenetic analysis in Figure 1, demonstrating that the *WRKY* transcription factors were highly conserved during evolution (Figure 3A). The number of conserved motifs contained in members of the *CcWRKY* gene family ranged from 3 to 9 (Figure 3B), in which motif 1 and 4 each contained a WRKYGQK conserved domain. All *CcWRKY* genes contained motif 1 and 2, and Group I also contained motif 4, which was consistent with the classification results of the *WRKY* gene family. Most *CcWRKY* ZFs comprised motifs 2 and 3, while the ZFs of *CcWRKY24*, *CcWRKY15*, *CcWRKY27*, *CcWRKY49*, *CcWRKY8*, and *CcWRKY13* in Group IIc were composed of motif 2 and 14. It was evident that the type, quantity, and distribution of conserved motifs within the same subgroup exhibited similarities. However, members of different groups only had access to specific conserved motifs. For instance, motif 15 was exclusive to members of Group III, whereas motif 4 was specific to Group I. These conserved motifs likely played a particular role in the *CcWRKY* gene family. According to exon–intron information, the *CcWRKY* genes exhibited a range of 1 (*CcWRKY44*)–7 (*CcWRKY53*) exons. Notably, approximately 53.4% of these genes contained three exons. Combined with evolutionary analysis, Group I genes mostly had four to six exons, with five exon genes accounting for 53.3% of Group I. In Group III, apart from *CcWRKY48* which contained four exons, others possessed three exons. Similarly, most genes in Group II had three exons (Figure 3C). To sum up, the conserved motifs and gene structures of *CcWRKY* genes within the same group were basically consistent, indicating that their evolutionary relationship was convergent, which also strongly verified the reliability of the population classification results.

### 2.4. Collinearity Analysis of WRKY Gene Family in C. chinense

Numerous gene replication events have been found in genome evolution. To clarify the replication event of *WRKYs*, we carried out the collinearity analysis of *WRKYs* in *C. chinense* and between *C. chinense* and *C. annuum*, *C. baccatum*, *Arabidopsis*, and *Oryza sativa*. The results showed that 7 gene pairs (*CcWRKY14/1*, *CcWRKY51/36*, *CcWRKY62/52*, *CcWRKY28/41*, *CcWRKY63/14*, *CcWRKY40/66*, and *CcWRKY36/41*) among 73 *WRKY* genes had a collinear relationship in *C. chinense*, respectively. Among them, only one gene replication occurred and five gene pairs were from Group II (Figure 4A). *C. chinense*, *C. annuum*, and *C. baccatum* belong to the Capsicum genus; there were 71 gene pairs between *C. chinense* and *C. annuum* (accounting for 95.9% of 74 *C. annuum WRKY* genes) and 57 gene pairs between *C. chinense* and *C. baccatum* (accounting for 78.1% of 73 *C. baccatum WRKY* genes), showing an apparent conservative collinear relationship (Figure 4B,C). Furthermore, 41 homologous genes were found in *C. chinense* and the dicotyledonous model plant *Arabidopsis*. In contrast, *C. chinense* had lower collinearity with *Oryza sativa*, with only 7 gene pairs of genes (accounting for 6.8% of 103 *OsWRKY* genes). Among them, *CcWRKY14*, *CcWRKY41*, *CcWRKY1*, *CcWRKY11*, *CcWRKY37*, *CcWRKY54*, and *CcWRKY23* genes had gene duplication events in five species, suggesting that these seven genes may be commonly found in dicotyledonous and monocotyledonous plants with a longer evolutionary history (Figure 4D,E).

### 2.5. Screening for Spiciness-Associated CcWRKY Transcription Factors

We used two sources of published transcriptome data to analyze the expression patterns of *CcWRKY* genes in different tissues of pepper and fruit development stages to understand the possibility of this gene participating in the regulation of pepper spiciness. Except for *CA12g19100* and *CA01g01900*—that exhibited exclusive expression in fruit—we discovered that the remaining *CcWRKYs* were expressed in at least two different types of pepper tissues, with significant variation in expression between tissues and developmental stages. Regarding the evolutionary tree findings, 11 genes (accounting for 68.8%) in Group I reached the peak expression levels between the young fruit stage (Fruits-Dev1) and the breaking stage (Fruits-Dev6). In contrast, the expression of the *CA07g10930* and *CA06g13580* genes was the highest 7 days after the breaking stage (Fruits-Dev9) (Figure 5A,B). Based on these findings, we speculated that the genes of Group I were likely involved in the regulation of chili fruit ripening and metabolite synthesis. Additionally, it could be seen that the expression of the *CA06g13580* gene in the placenta of pepper fruit 10 days after anthesis was relatively low, but its accumulation increased dramatically from 30 to 40 d and peaked at 50 d (Figure 5B,C and Figure 6B,C), showing a similar trend with the accumulation level of capsaicinoid (Figure 6A). It was speculated that the *CA06g13580* gene could participate in the biosynthesis of capsaicinoid.

### 2.6. Cloning and Subcellular Localization of CcWRKY25

The *CA06g13580* gene was 1750 bp, with an ORF of 1647 bp, encoding 548 amino acids, and isolated from Huangdenglong pepper (*C. chinense*) by RT-PCR. Since the sequence was similar to *AtMYB25*, it was subsequently renamed *CcWRKY25*. *CcWRKY25* was examined using qRT-PCR, revealing an increase in expression levels correlated with the pungency of the placenta in ripe pepper fruits. Notably, the expression of *CcWRKY25* was significantly higher in the extremely spicy Huangdenglong pepper (*C. chinense*) compared with the non-spicy Meiguoyuanjiao (*C. annuum*) (Figure 7A,B).

The results of the subcellular localization of *CcWRKY25* demonstrated that the fluorescent signal of pCAMBIA1300: GFP spread throughout the entire cell, in contrast to pCAMBIA1300: GFP: *CcWRKY25*, which had fluorescent signals only in the nucleus, consistent with the location prediction outcomes (Figure 7C).

### 2.7. Heterologous Expression of CcWRKY25

*CcWRKY25* successfully achieved heterologous transformation in *Arabidopsis* (Colombian wild type, WT). The expression of *PAL*, *4CL1*, *4CL2*, and *4CL3* were notably increased in *CcWRKY25*-overexpressing transgenic *Arabidopsis*. Notably, the structural genes involved in flavonoid biosynthesis, such as *F3H*, *FLS*, and *CHI*, exhibited significant upregulation. However, the expression of *CHS* was significantly decreased. Interestingly, genes related to lignin synthesis, including *C3H*, *CAD*, *COMT*, and *HCT*, showed downregulation, while *CCR* and *CCoAOMT* showed significantly higher expression levels in *CcWRKY25*-overexpressing transgenic *Arabidopsis* (Figure 8). It demonstrated that the overexpression of the *CcWRKY25* in *Arabidopsis* could change the structural genes of the lignin synthesis pathway and flavonoid biosynthetic pathway in varying degrees.

The flavonoid and lignin concentrations in transgenic and WT *Arabidopsis* were further evaluated to study the impact of heterologous expression of *CcWRKY25*. Compared with WT, the lignin content in *CcWRKY25*-overexpressing transgenic *Arabidopsis* significantly increased by 40%, while the flavonoid content showed a modest increase of only 4% (Figure 9F,G). Additionally, the transgenic plants were more robust and taller when comparing their growth with that of WT (Figure 9A–E). Intriguingly, the transgenic *Arabidopsis* plants exhibited delayed collapse, occurring approximately 7 weeks later than the WT plants. This indicated that *CcWRKY25* was likely to affect the phenotype of *Arabidopsis* by regulating lignin biosynthesis.

### 2.8. Effects of Silencing CcWRKY25 on the Biosynthesis of Capsaicinoids and Lignin

In order to further validate the function of *CcWRKY25* in the regulation of lignin and capsaicin biosynthesis, we realized the silencing of *CcWRKY25* in Huangdenglong pepper through VIGS. The expression of *CcWRKY25* was reduced by 54% in silenced peppers compared with uninfected peppers, whereas there was no discernible difference between uninfected peppers and pTRV2: empty peppers. Meanwhile, the expression of CBGs, including *PAL*, *C4H*, *4CL*, *pAMT*, *AT3*, and *KAS*, were reduced by 79%, 58%, 62%, 90%, 59%, and 77%, respectively. Compared with plants infected with pTRV2: empty, the transcription of *COMT* and *CCR* decreased in the silenced plants, but there was no noticeable difference (Figure 10).

The capsaicin, dihydrocapsaicin, and lignin in the placenta of fruits from different treatment groups at 45 days after anthesis were measured using HPLC methods. The findings demonstrated that the amounts of capsaicin and dihydrocapsaicin in the fruits of the pTRV2: empty plants and the uninfected plants were slightly reduced. In contrast, the lignin content in the fruit placenta of pTRV2: *CcWRKY25*-infected plants showed a 7.6% increase compared with uninfected plants. Interestingly, there was a significant decrease in the amounts of dihydrocapsaicin (25.6% reduction) and capsaicin (49.2% reduction) in the infected plants (Figure 11). It was speculated that *CcWRKY25*, by activating the expression of CBGs, could enhance the biosynthesis of capsaicinoid substances. Additionally, capsaicinoids and lignin could have a certain competitive relationship.

## 3. Discussion

### 3.1. WRKY Transcription Factors in C. chinense

The *WRKY* gene family is one of the most prominent families of transcriptional regulators in higher plants, playing crucial roles in physiological processes such as plant growth and response to biotic and abiotic stresses. *WRKYs* have been studied for more than 20 years and an increasing number of *WRKYs* have been reported in a variety of plants [14,28,29]. With genome publication [27], genome-wide analysis and functional characterization of *WRKY* in *C. chinense* has become possible. A total of 73 *CcWRKY* genes were discovered in the *C. chinense* genome, while there were 74, 109, 81, and 116 *WRKY* gene family members in *Arabidopsis* [30], *Oryza sativa* [15], *Solanum lycopersicum* [16], and *Gossypium* [29]. Along with a species genome, the abundance of *WRKYs* was also influenced by environmental factors to which the plants were exposed during their evolutionary process. Consequently, it was hypothesized that the *CcWRKY* gene family could have undergone some external environmental selection throughout the course of its natural development.

The *CcWRKY* gene family was divided into three groups: I, II (IIa, IIb, IIc, IId, IIe), and III, with the number of transcription factors in each group being 16 (21.9%), 47 (64.4%), and 10 (13.7%). However, in *Arabidopsis*, Group I members accounted for 42% of *AtWRKYs*, while Group II members accounted for 40% [30]. It was hypothesized that members of Group II went through more gene duplication processes during evolution. Gene segment duplication or self-replication was crucial for the growth and evolution of different gene family members [31]. Gene duplication events in *WRKY* genes were also found in *Arabidopsis* [30], *Solanum lycopersicum* [16], and *Oryza sativa* [32]. In *Solanum lycopersicum*, gene duplication was primarily associated with Group II members [16]; a similar trend was observed in *C. chinense*, which could explain the significant proportion of Group II members in *CcWRKYs*.

It was well established that the zinc finger motifs and WRKY domain of the *WRKY* gene family were highly conserved. However, there were some exceptions in the *CcWRKY* gene family. For instance, *CcWRKY49*, *CcWRKY47*, and *CcWRKY24* in Group IIe, as well as *CcWRKY63* in Group III, lost their zinc finger motifs. Previous studies indicated that Group I members in plants underwent mutations or losses in the C- and N-terminal WRKY domains or zinc finger motifs, thus evolving into Group II and Group III. Notably, Group III was exclusively present in higher plants [33]. Therefore, the loss of some Group IIe and Group III members’ zinc finger motifs could have facilitated the expansion of the *WRKY* gene family in *C. chinense*. Moreover, some CcWRKY proteins exhibited mutations in the highly conserved WRKY domain. For instance, WRKYGQK mutated to WRKYGKK (*CcWRKY64*, *CcWRKY67*, and *CcWRKY68*), WKRYGHK (*CcWRKY26*), WKRYGMK (*CcWRKY44*), and WKRYGQT (*CcWRKY31*). Similar variants have been observed in other species, such as *Nicotiana tabacum* [34], *Scutellaria baicalensis* [35], and *Caragana intermedia* [36]. These alterations in the WRKY domain could potentially impact the normal interaction of *WRKY* genes with their downstream target genes. For example, the mutation of the conserved WRKY motif (WRKYGQK) in *NtWRKY12* of *Nicotiana tabacum* to WRKYGKK initially led to specific binding to the W-box (TTGACT/C) of downstream genes; however, it then began binding to TTTTCCAC, resulting in the generation of a new function [34]. Consequently, the *CcWRKY* genes could endow new biological functions due to the zinc finger motif’s removal and the WRKYGQK motif’s mutation.

### 3.2. Functional Characterization of CcWRKY25

Capsaicinoid biosynthesis was tissue-specific and spatiotemporally specific, with synthesis initiating 15 days after anthesis and reaching its peak content between 40 and 50 days after anthesis. Subsequently, the capsaicin content gradually declined, exhibiting an “S” trend [8,9]. During this process, the expression of the *CcWRKY25* gene gradually increased in the placenta, reaching its peak at 50 days after anthesis, which aligned with previous findings [37,38]. Remarkably, we observed that the expression of *CcWRKY25* correlated with the level of pungency, suggesting its potential role as a regulator in capsaicinoid biosynthesis.

The phenylpropanoid pathway is a crucial pathway for the synthesis of secondary metabolites in plants. In this study, *CcWRKY25*-overexpressing transgenic *Arabidopsis* plants were grown taller with stronger stems. qRT-PCR analysis revealed a significant increase in the transcription levels of genes involved in lignin biosynthesis in the *CcWRKY25*-overexpressing transgenic *Arabidopsis* plants. Correspondingly, there was a notable increase in lignin content. Furthermore, the transcription of genes related to flavonoid synthesis, such as *F3H*, *FLS*, and *CHI*, was significantly upregulated upon *CcWRKY25*-overexpression; however, no significant changes in flavonoid content were observed. It was worth noting that CHS was considered to be the first rate-limiting enzyme for the 4-coumaric-CoA-catalyzed production of anthocyanidin [39]. Surprisingly, the overexpression of *CcWRKY25* in *Arabidopsis* significantly reduced *CHS* transcription, resulting in the reduced entry of 4-coumaric-CoA into the flavonoid pathway. Previously, it was reported that lignin and flavonoids competed for common precursors, such as 4-coumaric acid, cinnamic acid, ferulic acid CoA, and 4-coumaric CoA, to mutually inhibit their accumulations [40,41]. The overexpression of the *CcWRKY25* gene in *Arabidopsis* could divert additional intermediates towards the pathway, leading to lignin biosynthesis, which could explain the limited increase in flavonoid accumulation. However, the hypothesis mentioned above has to be confirmed.

*WRKY9* activated the transcription of *MYB31* by binding to the promoter W-box of *MYB31* to increase the content of capsaicin, confirming the regulatory role of *WRKY* in capsaicin synthesis [12,13]. In our study, the silencing of *CcWRKY25* significantly suppressed the transcription of the upstream genes, including *PAL*, *C4H*, and *4CL*, involved in phenylpropane biosynthesis, as well as CBGs such as *pAMT*, *AT3*, and *KAS*. Correspondingly, the accumulation of capsaicinoids was also significantly reduced. Notably, an enzyme (CS) for the last step of capsaicinoid biosynthesis was considered to be encoded by the *AT3* gene [42]. The expression patterns of *KAS*, *COMT*, *pAMT*, *BCAT*, *C4H*, *PAL*, and *4CL* genes were consistent with the trends observed in capsaicinoid synthesis [6,43], similar to our findings, with the difference that *COMT* expression was increased. COMT, being an enzyme with high substrate diversity, played a crucial role in the phenylpropanoid pathway and was involved in lignin synthesis. It should be noted that the silencing of peppers resulted in the decreased expression of *CAD* and *COMT* genes, which could alter the ratio of lignin monomers without affecting the overall lignin content [44,45]. In summary, we hypothesized that *CcWRKY25* could have a function in regulating capsaicinoid biosynthesis.

## 4. Materials and Methods

### 4.1. Plant Materials

Huangdenglong pepper (*C. chinense*), Wenchang xiaomijiao (*C. frutescens*), and Meiguoyuanjiao (*C. annuum*) were kept by the School of Horticulture, Hainan University. The seeds were germinated and sown in 32-hole seedling trays with the substrate (nutrient soil:vermiculite:perlite = 2:1:1) and cultivated under 28 °C: 16 h light/22 °C: 8 h darkness until about 6 leaves were transplanted into 5 L pots. Huangdenglong pepper pericarp and placenta were collected between 10 and 50 days after anthesis to assess the synthesis trend of capsaicinoids and to further validate gene expression. The placenta Huangdenglong pepper (*C. chinense*), Wenchang xiaomijiao (*C. frutescens*), and Meiguoyuanjiao (*C. annuum*) were collected at fruit breaker (about 35 days after anthesis) for qRT-PCR.

The School of Tropical Crops, Hainan University, provided seeds of *Arabidopsis* (Colombian wild type). After vernalization at 4 °C for 2 days, the seeds were evenly spread on the substrate. When the cotyledons had fully developed, they were transplanted onto seedling trays under the cultural conditions of (24 °C: 16 h/18 °C: 8 h and 85% humidity). The transformation was performed when they began to branch out and form flowering inflorescences (about 5 weeks).

Seeds of *Nicotiana benthamiana* were sown in cavity trays to germinate. When the cotyledons were fully expanded, the tobacco was transplanted into seedling pots and placed in an incubator at 26 °C light/20 °C dark and 85% humidity. After 4 weeks, until the leaves were 6–8 per plant, they were used for subcellular localization tests.

### 4.2. Bioinformatic Analysis of CcWRKY Gene Family

The *C. chinense* PI159236 genome and the WRKY domain (PF03106) were retrieved from the Pepper Genome Database (PGD, http://pgd.pepper.snu.ac.kr/, accessed on 28 June 2022) and the Pfam database, respectively. The WRKY domain was utilized as a search reference in the *C. chinense* genome using the HMMER; the screening condition was EValue < 0.001 to obtain the initial *WRKY* gene family members. The protein sequences of these candidate genes were obtained from gene IDs and submitted to websites such as SMART and Pfam for verification, in which sequences without complete WRKY domains were excluded. Physicochemical and chemical properties of CcWRKY proteins, such as pI, molecular weight, and hydrophilicity/hydrophobicity, were predicted via ExPASy. The subcellular localization was carried out by PSORT II.

The protein sequences of the *Arabidopsis WRKY* gene family were retrieved from the Arabidopsis Information Resource (TAIR, https://www.arabidopsis.org/, accessed on 28 June 2022) to investigate the evolutionary relationship of the *WRKY* gene family between *C. chinense* and *Arabidopsis*. MEGA X was used to create a phylogenetic tree from the protein sequences of CcWRKYs and AtWRKYs by the neighbor-joining method (NJ). The Evolview website was used to visualize the evolutionary tree, Jalview software (2.11.2.0) was used to beautify the multiple sequences, and the seqlogo was created using the WebLogo 3.

The MEME online website predicted the conserved motifs of the *WRKY* genes with the motif number set to 20. The structural domains of the *CcWRKY* genes were analyzed by the online website CDD. For further analysis, the gene structure, conserved motifs, and phylogenetics of *CcWRKYs* were displayed via TBtools.

A blast comparison of *WRKY* genes in *C. chinense* was performed and MC-ScanX was utilized to search for collinearity between *CcWRKY* genes, which was visualized using Circos-0.69 software. To further analyze the collinearity analysis of WRKY proteins, the genome data of *C. annuum*, *C. baccatum*, *Arabidopsis*, and *Oryza sativa* were downloaded separately from the Ensemble database.

### 4.3. Expression Patterns of CcWRKYs

Illumina RNA-seq datasets were downloaded from PGD to examine the expression patterns of *CcWRKY* genes in various parts of the pepper, including root, stem, leaf, bud, flower, and fruit, as well as the placenta of the pepper at 10–50 days after anthesis. After normalizing the data by log2(TPM+1), TBtools software produced a heat map.

### 4.4. RNA Extraction and RT-qPCR Analysis

The total RNA was extracted from samples of pepper and *Arabidopsis* by an RNA extraction kit (Novozymes, Nanjing, China) and measured using a spectrophotometer (Thermo Scientific, Wilmington, DE, USA), then reverse-transcribed to cDNA using HiScript III 1st strand cDNA synthesis kit (Novozymes, Nanjing, China). Primer sequences were designed using Premier 5.0 software (Appendix A). As housekeeping genes, *CcActin* (accession: AY486137.1) from *C. chinense* and *AtActin* (accession: NM_001338359.1) from *Arabidopsis* were utilized. With the aid of the 2ChamQ Universal SYBR qPCR master mix (Novozymes, Nanjing, China), RT-qPCR reactions were carried out and the 2^−ΔΔCT^ method was ultimately used to process the data.

### 4.5. Cloning and Subcellular Localization Assay of CcWRKY25

*CA06g13580* was cloned from Huangdenglong pepper using *CcWRKY25*-F/R primers (Appendix A). The coding sequence (CDS) without stop codon of *CcWRKY25* was inserted into the cloning vector TA before subcloning into the pCAMBIA1300: GFP vector. The pCAMBIA1300: GFP: *CcWRKY25* fusion protein and pCAMBIA1300: GFP were separately and transiently synthesized in tobacco leaves after *Agrobacterium* infection [45]. The infective tobacco was cultured in a light-proof environment at 24 °C for 2 days. Then, fluorescence was detected by laser confocal microscopy (Nikon, Japan).

### 4.6. CcWRKY25 Transformation in Arabidopsis

The CDS of *CcWRKY25* was used to construct the pBI121: *CcWRKY25* vector. The floral dip method transformed the empty vector (pBI121: empty) and pBI121:*CcWRKY25* into *Arabidopsis*, as described by Ali et al. [46]. The T0 generation *Arabidopsis* seeds were planted in 1/2MS medium containing 50 g/mL kanamycin (Kan), and positive seedlings with 2–4 green leaves were transplanted in soil. After about one month, RNA was extracted from T1-generation transgenic plants and WT and verified by semi-quantitative RT-PCR using pB: *CcWRKY25*-F/R primers (Appendix A). The transgenic *Arabidopsis* plants were then bred and screened until pure-hybrid T2 generation lines were obtained. The transcription of the *CcWRKY25* in T2 generation transgenic *Arabidopsis* plants was examined by RT-qPCR (Appendix A); three T2 generation lines (W25-2, W25-3, and W25-4) with high *CcWRKY25* expression were selected for phenotypic observation and further analysis. *Arabidopsis* were cultured for about 6 weeks to measure the diameter of the main stem and height.

### 4.7. Silencing the CcWRKY25 Gene in Huangdenglong Pepper

A specific fragment of the CDS in the *CcWRKY25* gene, ranging from 300 to 500 bp, was selected and inserted into the pTRV2 vector. The pepper plants were divided into three groups: a blank control group without treatment (uninfected), a positive control group infected with a mixture of pTRV1 and pTRV2: empty (pTRV2: empty), a test group infected with a mixture of pTRV1 and pTRV2: *CcWRKY25* (pTRV2: *CcWRKY25*). Inoculation was performed concerning the method described by Kim et al. [47].

Each treatment involved the infection of five pepper plants. Treated peppers were grown in the darkness at 20 °C for 2 days. After 2 days, the culture condition was adjusted to 24 °C (16 h: light) and 22 °C (8 h: darkness). Pepper fruits from various treatments were harvested 45 days after anthesis to assay.

### 4.8. Measurement of Lignin, Flavonoids, and Capsaicinoid Content

The lignin and flavonoid contents were measured by a Solarbio lignin content detection kit (Solarbio, Beijing, China) and Nanjing Jiancheng Bioengineering Institute (Nanjing, China), respectively. To evaluate the capsaicin (CAP) and dihydrocapsaicin (DhCAP) contents in the placenta of the pepper fruits from the corresponding samples, CAP and DhCAP were extracted and quantified by the HPLC method [48]. In summary, capsaicin standard substance (purity ≥ 98%) of 5 mg and dihydrocapsaicin standard substance (purity ≥ 98%) of 5 mg were dissolved in 5 mL of methanol (HPLC-grade chromatographic alcohol) to prepare a standard solution with a concentration of 1.0 mg/mL. An amount of 1 g of the sample was mixed with 5 mL of methanol and subjected to 30 min of ultrasonic extraction. The standard substances and samples were separately placed in a high-performance liquid chromatography system (Agilent 1200, Agilent, NY, USA) equipped with a chromatographic column (4.6 × 250 mm, 5 μm, Waters Symmetry^®^ C18, Waters, Milford, MA, USA). The mobile phase consisted of methanol (80% *v*/*v*) and ultrapure water (20% *v*/*v*), with a flow rate of 1.0 mL/min. Detection was performed at a UV wavelength of 280 nm.

### 4.9. Statistical Analysis

All experiments were conducted in triplicate and the mean standard deviation was calculated. Statistical analysis was performed using SPSS 25 software, based on one-way ANOVA, followed by a Tukey–Kramer post hoc test (*p* ≤ 0.05) for statistically significant differences. GraphPad Prism 9.0 software was used for graphing.

## 5. Conclusions

We identified a total of 73 *WRKY* transcription factors in *C. chinense* and classified them according to the characteristics of the *WRKY* gene family and *AtWRKYs*. The results of qRT-PCR and transcriptome analysis suggested that *CcWRKY25* could serve as a potential regulator of capsaicinoid biosynthesis. *CcWRKY25* was a nuclear-localized transcription factor, whose expression level increased with spiciness, according to a series of experiments. The ectopic expression of *CcWRKY25* in *Arabidopsis* enhanced the expression of the phenylpropanoid pathway’s upstream genes, while also promoting the formation of lignin, giving transgenic plants more strength and height. Interestingly, silencing the CcWRKY25 gene significantly reduced the expression of CBGs and the accumulation of capsaicin. However, there was no significant difference in lignin content. In conclusion, our findings provided new insight into the regulatory mechanism of capsaicinoid biosynthesis and laid the groundwork for further comprehensive analysis of CcWRKY proteins.

## Figures and Tables

**Figure 1 ijms-24-11389-f001:**
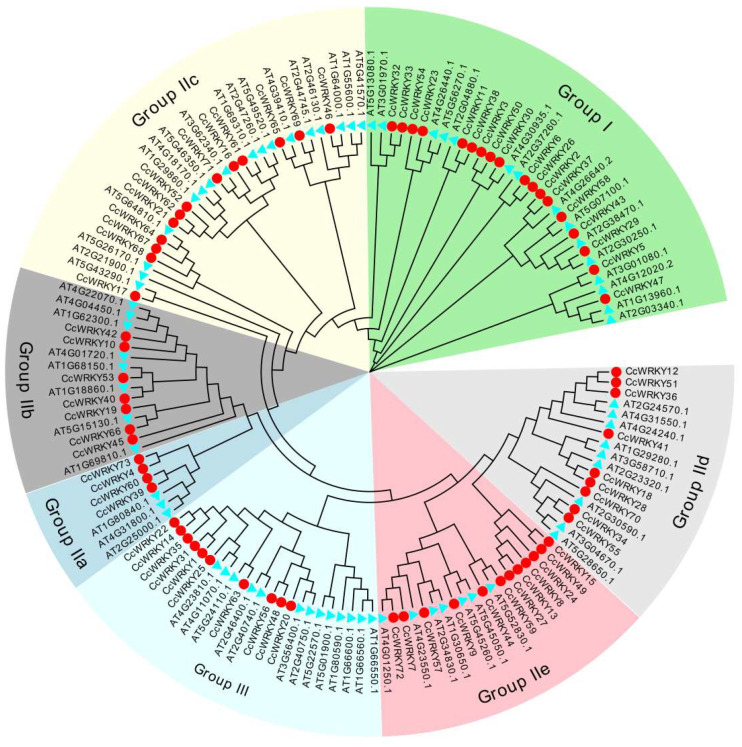
Phylogenetic tree of WRKY proteins from *C. chinense* (red circle) and *Arabidopsis thaliana* (light blue triangle). The WRKYs were divided into Groups I, II, and III; Group II can be further classified into five subgroups. Different groups and subgroups are distinguished using different color modules.

**Figure 2 ijms-24-11389-f002:**
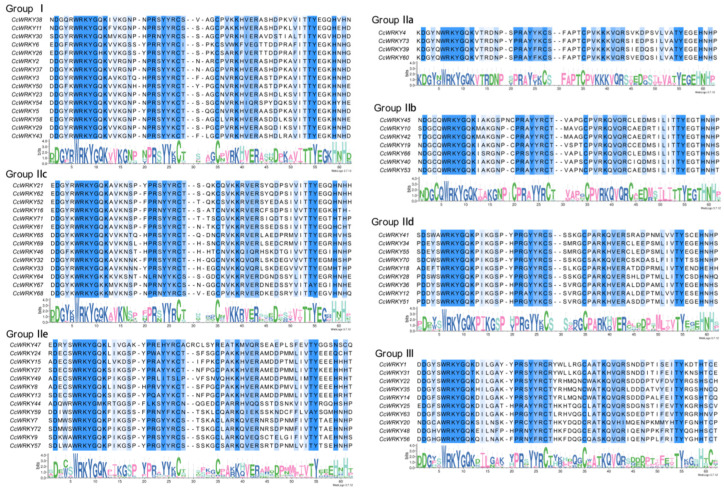
Multiple sequences alignment of the conservative domain from CcWRKY proteins. The conserved WRKY domains and the zinc finger motif are marked in deep blue.

**Figure 3 ijms-24-11389-f003:**
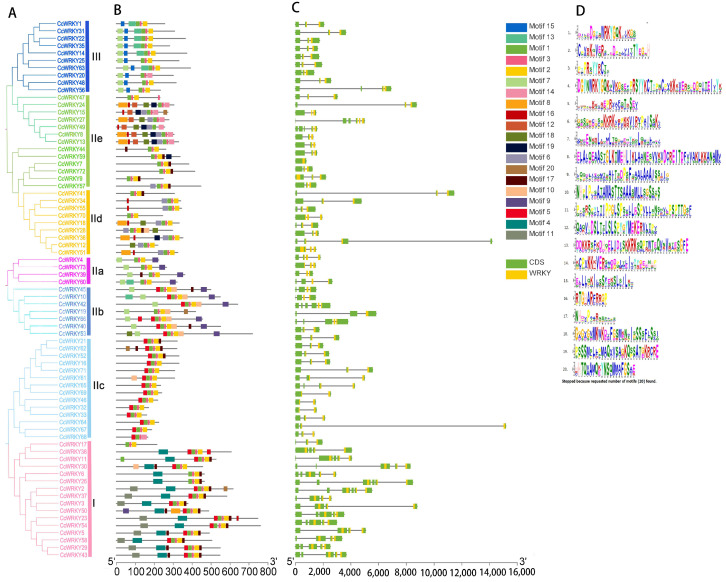
Analysis of CcWRKY conserved motifs and gene structure. (**A**) CcWRKY protein phylogenetic tree, different colors were used to distinguish groups and subgroups. (**B**) Conserved motif of 73 CcWRKY proteins. (**C**) Gene structure of CcWRKYs. (**D**) The amino acid sequences of 20 motifs of CcWRKY proteins.

**Figure 4 ijms-24-11389-f004:**
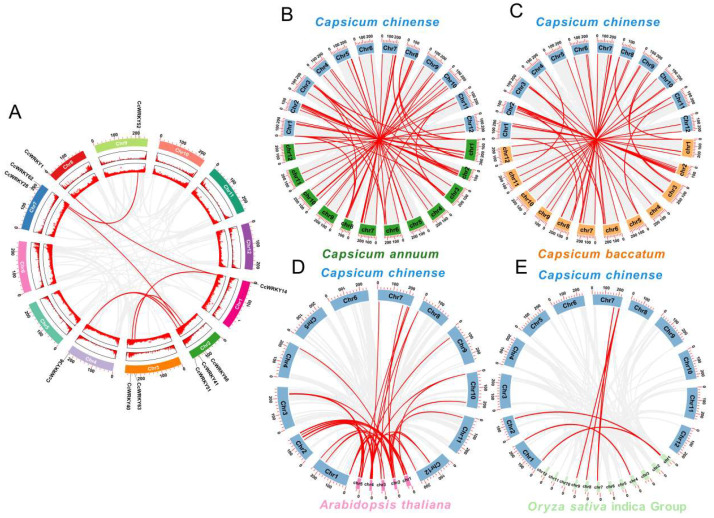
Collinearity analysis of *WRKY* genes. (**A**) Segmentally duplicated gene pairs in the *C. chinense* genome. (**B**–**E**) Synteny analyses of *C. chinense WRKY* with *C. annuum*, *C. baccatum*, *Arabidopsis*, and *Oryza sativa* indica group. The collinear blocks generated by the *C. chinense* and other plant genomes are indicated with gray lines in the background, whereas syntenic *WRKY* gene pairs with the red lines.

**Figure 5 ijms-24-11389-f005:**
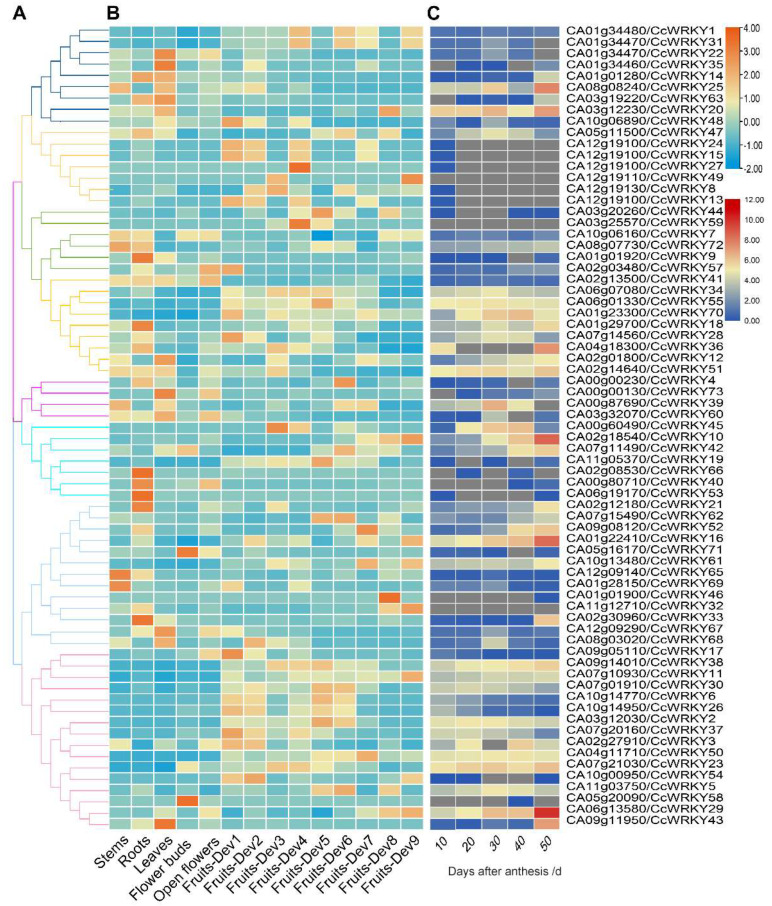
Cluster analysis of the *WRKY* gene expression profiles in *C. chinense*. (**A**) CcWRKY protein phylogenetic tree. (**B**) Cluster analysis was performed based on expression profiles of *CcWRKY* genes in various tissues (root, stem, leaf, flower, and fruit). (**C**) Expression profiles of *CcWRKYs* in placenta of pepper from 10 to 50 days after anthesis. The color boxes ranging from blue to red indicate an expression from low to high.

**Figure 6 ijms-24-11389-f006:**
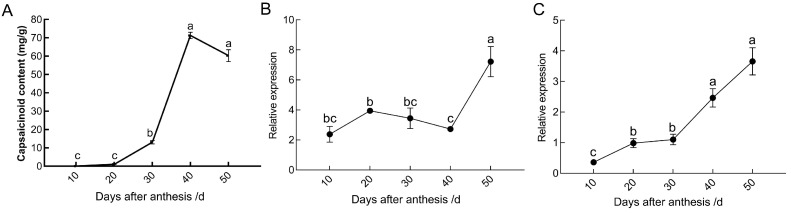
Screening for spiciness-associated *CcWRKY* transcription factors. (**A**) Capsaicinoid content of placenta from 10 to 50 days after anthesis. *CA07g10930* (**B**) and *CA06g13580* (**C**) expression of placenta from 10 to 50 days after anthesis. Data on the bars marked without the same lowercase letter indicate significant differences at *p* ≤ 0.05.

**Figure 7 ijms-24-11389-f007:**
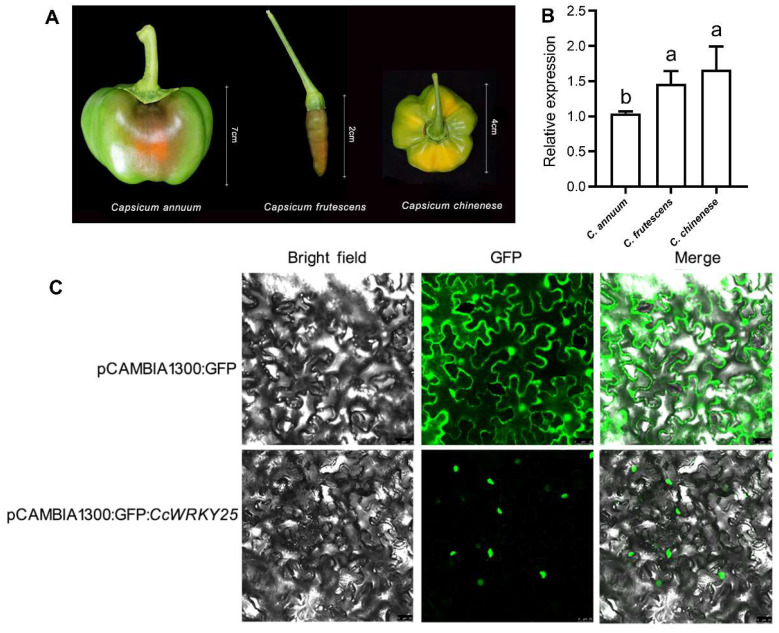
Relative expression and subcellular localization analysis of *CcWRKY25*. (**A**) The fruit of *C. annuum*, *C. frutescens*, and *C. chinense*. (**B**) Relative expression of *CcWRKY25* in placenta of fruit in different pungent cultivars at the breaker. (**C**) Subcellular localization analysis of *CcWRKY25*. Data on the bars marked without the same lowercase letter indicate significant differences at *p* ≤ 0.05.

**Figure 8 ijms-24-11389-f008:**
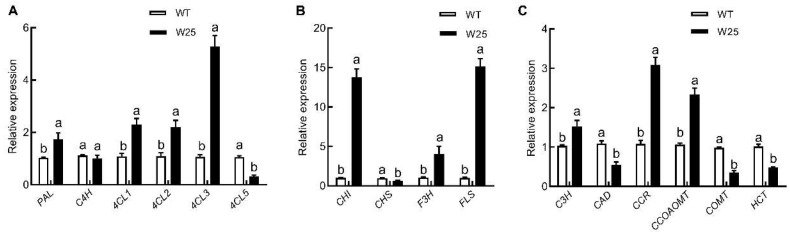
qRT-PCR analysis of genes related to phenylpropane (**A**), flavonoid (**B**), and lignin (**C**) anabolic pathways in W25 transgenic *Arabidopsis.* Data on the bars marked without the same lowercase letter indicate significant differences at *p* ≤ 0.05.

**Figure 9 ijms-24-11389-f009:**
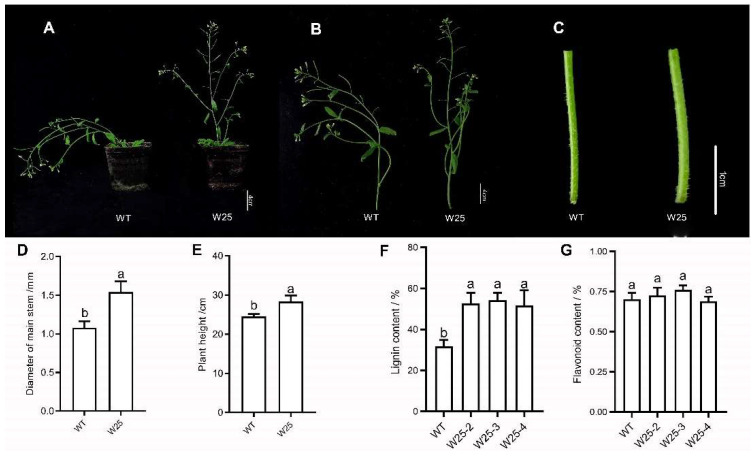
T2 generation transgenic *Arabidopsis*. (**A**) Whole plant of WT and W25. (**B**) Aerial part of WT and W25. (**C**) Detail of main stem of WT and W25. (**D**) Columnar diameter of main stem of WT and W25. (**E**) The height histogram of WT and W25. (**F**) The lignin contents in WT and W25. (**G**) The flavonoid contents in WT and W25. Data on the bars marked without the same lowercase letter indicate significant differences at *p* ≤ 0.05.

**Figure 10 ijms-24-11389-f010:**
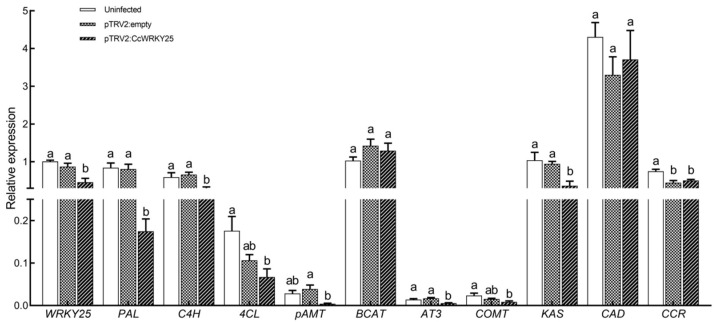
qRT-PCR analysis of *CcWRKY25* and CBGs in placenta of *C. chinense* 45 days after anthesis. Data on the bars marked without the same lowercase letter indicate significant differences at *p* ≤ 0.05.

**Figure 11 ijms-24-11389-f011:**
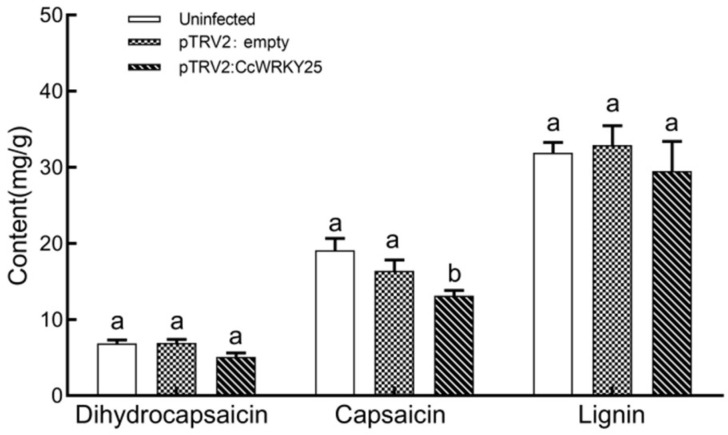
Dihydrocapsaicin, capsaicin, and lignin contents in the placenta of different treatment groups of *C. chinense* at 45 days after anthesis. Data on the bars marked without the same lowercase letter indicate significant differences at *p* ≤ 0.05.

## Data Availability

The data and materials that support the findings of this study are available from the corresponding author upon reasonable request.

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
