# Peer review of "Genome-Wide Identification of WRKY Gene Family and Functional Characterization of CcWRKY25 in Capsicum chinense"

_ijms, 2023, doi:10.3390/ijms241411389_

Round 1
Reviewer 1 Report
The manuscript entitled “Genome-wide identification of WRKY gene family and functional characterization of CcWRKY25 in Capsicum chinense” deals with the role of WRKY transcription factors in the synthesis of capsaicin. The work objectives are clear, the experimental design is correct, results and discussion are addressed properly. The redaction is good, so the manuscript is easy to follow in spite of the many methodologies employed. My main concern is related to some of the references cited on the main text, many of which do not support the affirmations done by authors. Some examples are references [1], [7], [26], [48], [37].
Other minor questions I would like to be solved by authors are exposed below:
I am not sure why authors employ the term “spicy” instead of “pungency”.
Line 40: “In the placenta of the plant?” Perhaps authors mean “in the placenta of the fruit”?
Line 65: “with 74, 109, 81, and 164, respectively”. What those numbers are?
Line 378 and 381: “10-50 days after breaking” “fruit breaker”. What is the meaning of “breaking” and “fruit breaker”?
Line 393: “from the pepper genome database”. Could you please specify which database was employed?
Line 403: Define TAIR
Line 418: Define PGD
What was the housekeeping gene in qPCR analysis?
Line 470: “were extracted and quantified using the method by following the national measurement standard [48]”. What is the national measurement standard?
Line 481: What kind of statistical analysis were applied?
Line 98: “C. chinense’genome”. What C. chinense genome was used?
Figure 1: Phylogenetic tree of WRKY proteins from C. chinense(●) and Arabidopsis thaliana (▲). I cannot see those symbols on the figure. Instead, what is the meaning of blue and red circles?
Line 332: “displaying an “S” trend”. Could authors explained what is a S trend?
Author Response
We feel great thanks for your professional review work on our article. As you are concerned, there are several problems that need to be addressed. Taking into account your insightful suggestions, we have made the necessary corrections to our previous manuscript. The detailed revisions are outlined below.
Comment 1: Related to some of the references cited on the main text, many of which do not support the affirmations done by authors. Some examples are references [1], [7], [26], [48], [37].
Reply 1: We consider this to be an excellent suggestion. Capsicum consists of five cultivated species, as mentioned in reference [1]. In accordance with your advice, I have replaced the references with more relevant ones that align with the content of the article, such as references [7], [25], [26], [48]. Moreover, I apologize for our oversight that led to a mismatch between the content cited in the paper and the corresponding references starting from [28]. We have rectified this error by making the necessary corrections and highlighting all the revised references in red. Once again, I sincerely regret our carelessness and appreciate your valuable reminder.
Comment 2: I am not sure why authors employ the term “spicy” instead of “pungency”.
Reply 2: Thank you for your comment. "spicy" is an adjective, while "spiciness" is its corresponding noun form. Similarly, "pungent" is an adjective , and "pungency" is a noun. I have reviewed the relevant literature, and in fact, both terms can be used. Pungency is often associated with the presence of chemical compounds such as capsaicin. Pungency is not limited to spicy or hot flavors, as it can also be present in foods like mustard, horseradish, or certain types of onions. On the other hand, spiciness refers specifically to the perception of heat or a burning sensation caused by the presence of compounds like capsaicin in chili peppers. Thank you again for considering the word suggestions. In order to diversify the vocabulary, I have replaced some words of "spiciness" with "pungency."
Comment 3: Line 40: “In the placenta of the plant?” Perhaps authors mean “in the placenta of the fruit”?
Reply 3: Yes, I would like to express "in the placenta of the fruit." Thank you for your reminder. I have made the necessary replacements in line 40.
Comment 4: Line 65: “with 74, 109, 81, and 164, respectively”. What those numbers are?
Reply 4: Thank you for your comment. I apologize for the confusion caused by my wording. This is the original sentence from the manuscript (line 63-65):"WRKY, one of the most prominent transcription factor families in higher plants, has been identified in Arabidopsis [14], Oryza sativa [15], Solanum lycopersicum [16], and Nicotiana Tabacium [17] with 74, 109, 81, and 164, respectively." It means that 74 WRKY genes have been identified in Arabidopsis, 109 WRKY genes in Oryza sativa, 81 WRKY genes in Solanum lycopersicum, and 164 WRKY genes in Nicotiana Tabacium.
Comment 5: Line 378 and 381: “10-50 days after breaking” “fruit breaker”. What is the meaning of “breaking” and “fruit breaker”?
Reply 5: We sincerely thank the reviewer for careful reading and we were really sorry for our careless mistakes. “Fruit breaker” is indeed recognized as the period when the fruit's color begins to change, such as from green (immature) to red (ripe). In our paper, the phrase "10-50 days after breaking" was used incorrectly. It should be "10-50 days after anthesis." In the revised manuscript, I have replaced "days after breaking" with "days after anthesis" in both the Figure 6 and the text.
Comment 6: Line 393: “from the pepper genome database”. Could you please specify which database was employed?
Reply 6: Thank you for your comment. Pepper Genome Database (PGD), can be accessed at http://pgd.pepper.snu.ac.kr/. In our resubmitted manuscript, the detail of pepper genome database has been referenced in line 395-396.
Comment 7: Line 403: Define TAIR
Reply 7: We sincerely thank the reviewer for careful reading. The Arabidopsis Information Resource (TAIR, https://www.arabidopsis.org/) have been modified in resubmitted manuscript (line 406).
Comment 8: Line 418: Define PGD
Reply 8: We sincerely thank the reviewer for careful reading. Pepper Genome Database (PGD) has been mentioned in the line 395-396, so the abbreviation will continue to be used in the line 422.
Comment 9: What was the housekeeping gene in qPCR analysis?
Reply 9: Thank you for your patient review. In the qPCR experiment, the housekeeping gene for Arabidopsis thaliana is AtActin (accession: NM_001338359.1), while for Capsicum chinense, it is CcActin (accession: AY486137.1). I have highlighted this information in lines 431-433.
Comment 10: Line 470: “were extracted and quantified using the method by following the national measurement standard [48]”. What is the national measurement standard?
Reply 10: Thank you for your feedback, and we apologize for any confusion caused by our description of the method. We have made revisions to the citations in the revised manuscript (line 472-474).
Comment 11: Line 481: What kind of statistical analysis were applied?
Reply 11: We sincerely thank the reviewer for careful reading. In this paper, we use one-way ANOVA followed by Tukey-Kramer post-hoc test to perform the statistical analysis. In our resubmitted manuscript, I have listed the test in line 486-488.
Comment 12: Line 98: “C. chinense’genome”. What C. chinense genome was used?
Reply 12: Thank you for your comment. We used the C. chinense PI159236 genome, and the reference for the genome has been added to the resubmitted manuscript in the line 395-396.
Comment 13: Figure 1: Phylogenetic tree of WRKY proteins from C. chinense(●) and Arabidopsis thaliana (▲). I cannot see those symbols on the figure. Instead, what is the meaning of blue and red circles?
Reply 13: Thank you for your careful comment. The symbol "●" is the color red, while "▲" is the color blue. These symbols are positioned between the phylogenetic tree and the names. As there are three group and five subgroups within the WRKY gene, for better visibility, I have marked the different groups and subgroups using different colors. For example, in the Figure 1, WRKY group I is represented by the green module.
Comment 14: Line 332: “displaying an “S” trend”. Could authors explained what is a S trend?
Reply 14: Thank you for your careful comment. Capsaicin gradually accumulates in pepper fruits starting from 16 days after anthesis (DAA) and reaches its peak content between 40-50 DAA. Subsequently, the capsaicin content gradually declines, exhibiting an "S" trend. Thank you once again for your suggestions. I have detailed the the "S" trend in the revised manuscript (line 332-334).
If there are any other modifications we could make, we would likevery much to modify them and we really appreciate your help.
Reviewer 2 Report
Here are some minor notes:
1) Figure 3D is hard to read, maybe it could be inserted in a better resolution?
2) In Figure 6-11 instead of P<0.05, it should be p<0.05.
3) There are no letters in Figure 6 to indicate statistically significant differences? They are in the description below the figure.
4) What tests were used to perform the statistical analysis? Parametric or non-parametric? Has an analysis of variance been performed? Please list specific tests in materials and methods?
5) I did not find a reference to Figure S2 in the main text either.
Author Response
Thank you very much for your careful review of our research paper, your professional insights and suggestions have played a crucial role in guiding our study, and we sincerely appreciate your support and patience. After incorporating your feedback, we have made detailed adjustments and improvements to the manuscript. We have carefully read each comment and seriously considered your recommendations. The following are the modifications we have made based on your suggestions.
Comment 1: Figure 3D is hard to read, maybe it could be inserted in a better resolution?
Reply 1: Thank you for your comment, we have rearranged the Figure 3 in the resubmitted manuscript, but it might be necessary to enlarge the image to clearly observe Figure 3D. I appreciate your understanding.
Comment 2: In Figure 6-11 instead of P<0.05, it should be p<0.05.
Reply 2:We sincerely thank the reviewer for careful reading. As suggested by the reviewer, we have corrected the “P<0.05” into “p≤0.05” in Figure 6-11 and line 487.
Comment 3: There are no letters in Figure 6 to indicate statistically significant differences? They are in the description below the figure.
Reply 3:Thanks for your correction. Since capsaicin is mainly synthesized in the placenta, we not only added the letter to Figure 6 in the revised manuscript, but also removed the expression of CA07g10930 and CA06g13580 in the pericarp. This way, the relationship between these two genes and capsaicin synthesis can be clearly observed.
Comment 4:What tests were used to perform the statistical analysis? Parametric or non-parametric? Has an analysis of variance been performed? Please list specific tests in materials and methods?
Reply 4: Thank you for your careful comment. In this paper, both parametric tests were employed for the statistical analysis. An analysis of one-way ANOVA followed by Tukey-Kramer post-hoc test was performed perform the statistical analysis. The details of test, including the specific variables analyzed and the significance thresholds applied, are outlined in the materials and methods section of the resubmitted manuscript (line 486-488).
Comment 5:I did not find a reference to Figure S2 in the main text either.
Reply 5:Thanks for your correction. We feel sorry for our carelessness. In our resubmitted manuscript, Figure S2 has been referenced in line 454.
If there are any other modifications we could make, we would likevery much to modify them and we really appreciate your help.